# Transient Activation of Hedgehog Signaling Inhibits Cellular Senescence and Inflammation in Radiated Swine Salivary Glands through Preserving Resident Macrophages

**DOI:** 10.3390/ijms222413493

**Published:** 2021-12-16

**Authors:** Liang Hu, Conglin Du, Zi Yang, Yang Yang, Zhao Zhu, Zhaochen Shan, Chunmei Zhang, Songlin Wang, Fei Liu

**Affiliations:** 1Salivary Gland Disease Center and Beijing Key Laboratory of Tooth Regeneration and Function Reconstruction, Beijing Laboratory of Oral Health, School of Stomatology, Capital Medical University, Beijing 100050, China; huliang1117@gmail.com (L.H.); duconglin@ccmu.edu.cn (C.D.); yzi19345@gmail.com (Z.Y.); yangsun910@ccmu.edu.cn (Y.Y.); zhuzhao420@gmail.com (Z.Z.); kqjy55@gmail.com (C.Z.); 2Outpatient Department of Oral and Maxillofacial Surgery, School of Stomatology, Capital Medical University, Beijing 100050, China; shanzhch629@gmail.com; 3Department of Molecular and Cellular Medicine, Institute for Regenerative Medicine, College of Medicine, Texas A&M University, College Station, TX 77843, USA

**Keywords:** salivary gland, hedgehog signaling, resident macrophages, cellular senescence, inflammation, radiation

## Abstract

Salivary gland function is commonly and irreversibly damaged by radiation therapy for head and neck cancer. This damage greatly decreases the patient’s quality of life and is difficult to remedy. Previously, we found that the transient activation of Hedgehog signaling alleviated salivary hypofunction after radiation in both mouse and pig models through the inhibition of radiation-induced cellular senescence that is mediated by resident macrophages in mouse submandibular glands. Here we report that in swine parotid glands sharing many features with humans, the Hedgehog receptor PTCH1 is mainly expressed in macrophages, and levels of PTCH1 and multiple macrophage markers are significantly decreased by radiation but recovered by transient Hedgehog activation. These parotid macrophages mainly express the M2 macrophage marker ARG1, while radiation promotes expression of pro-inflammatory cytokine that is reversed by transient Hedgehog activation. Hedgehog activation likely preserves parotid macrophages after radiation through inhibition of P53 signaling and consequent cellular senescence. Consistently, VEGF, an essential anti-senescence cytokine downstream of Hedgehog signaling, is significantly decreased by radiation but recovered by transient Hedgehog activation. These findings indicate that in the clinically-relevant swine model, transient Hedgehog activation restores the function of irradiated salivary glands through the recovery of resident macrophages and the consequent inhibition of cellular senescence and inflammation.

## 1. Introduction

Head and neck cancers (HNC) account for 3.6% of all cancers in the USA, comprising an estimated 65,630 new cases in 2020 [1] and an estimated 429,930 HCN survivors in 2030 [2]. Radiotherapy is a primary treatment for HNC, but it frequently exposes non-diseased salivary glands (SGs) to radiation therapy. Among the three pairs of major salivary glands and numerous minor salivary glands, submandibular glands produce approximately two-thirds of unstimulated saliva, while parotid glands produce approximately one-half of stimulated saliva. SGs are sensitive to radiation despite the slow proliferation of SG epithelial cells. In many HNC patients treated with radiation therapy, both unstimulated and stimulated saliva flow rates persistently decrease to barely measurable levels [3]. Such an irreversible salivary hypofunction is common in HNC survivors treated with either conventional radiotherapy (incidence > 60%) [3] or novel intensity-modulated radiotherapy (incidence ≈ 20%) [4]. The consequent long-term dry mouth severely exacerbates dental caries and periodontal disease and causes problems related to taste, sleep, and speech that severely impair the patient’s quality of life. Current treatments for radiation-induced salivary hypofunction, such as artificial saliva and saliva-secretion stimulators, can only provide partial and temporary relief since they are often of short duration, lack the protective effects of saliva, or have potentially significant adverse effects [5] The only preventive measure strongly recommended by the Multinational Association of Supportive Care in Cancer/International Society of Oral Oncology/American Society of Clinical Oncology (MASCC/ISOO/ASCO) is limiting the cumulative dose and the irradiated volume of SGs [6], which might be infeasible for some HNC patients due to cost or efficacy concerns.

Cellular senescence and related persistent inflammation are key causes of irreversible dry mouth after radiation [7,8]. In a mouse model, we found that the transient activation of Hedgehog signaling restored saliva secretion using multiple mechanisms, including the inhibition of cellular senescence-caused inflammation [9,10,11], which is mainly mediated by the Hedgehog-responsive resident macrophages (rMϕs) [12]. In the steady-state rodent SGs, rMϕs are the most abundant immune cells (>80%) [13,14], extend numerous dendrites throughout SGs, and attach to neighboring epithelial or endothelial cells [15,16]. In salivary glands damaged by duct ligation, macrophages are the dominant phagocytes for clearing dead or senescent cells [17]. In adult mouse submandibular glands (SMGs), resident macrophages interact with epithelial progenitors and endothelial cells through homeostatic factors, including complement component 1q (C1q) and hepatocyte growth factor (Hgf) [12]. Radiation sharply and persistently decreases the number of SMG resident macrophages and their homeostatic interactions with other cells, but this process is reversed by the transfer of the Sonic Hedgehog (Shh) gene [12]. 

Human salivary glands differ from rodent salivary glands but are much more comparable to those found in pigs in many features, including size, morphology, and gene-expression profiles [18,19,20]. The pig parotid model of radiation-induced salivary gland hypofunction has been well established and is required by the FDA as the proof-of-concept prior to the initiation of clinical trials [21,22]. We have confirmed that initiating transient Hedgehog activation in the parotids of pigs 4 weeks after radiation therapy restored salivary function for up to 20 weeks [23]. The underlying mechanisms are related to the alleviation of radiation damage on microvessels and the inhibition of radiation-induced cellular senescence [23]. However, the exact cellular and molecular mechanisms underlying this rescue effect in the pig model remain unclear. Here we report that macrophages in swine parotid glands are major Hedgehog-responsive cells and are essential for salivary gland function through regulating cellular senescence and inflammation. These findings indicate that macrophages are promising targets for alleviating radiation damage to salivary glands and potentially other healthy organs as well.

## 2. Results

### 2.1. Hedgehog Signaling in Pig Parotids Is Inhibited by Radiation and Restored by Shh Gene Transfer with the Preservation of Salivary Gland Function

We have reported that the intra-parotid transfer of the Shh gene effectively mitigated the detrimental effect of radiation on the function of swine salivary glands [23]. However, due to the lack of specific antibodies for swine antigens, we have not determined the effects of radiation with or without Shh gene transfer on Aquaporin 5 (AQP5), the water-channel protein essential for saliva secretion and Hedgehog signaling activity at the protein level. Recently, many more antibodies recognizing swine antigens have been developed. Therefore, we repeated the parotid radiation and the transfer of the Shh or control GFP gene at 4 weeks after radiation in the miniature pig model [23], measured pilocarpine-stimulated parotid saliva flow rates at 20 weeks after radiation, and collected parotids at 1, 5, and 20 weeks after radiation (Figure 1A). At 20 weeks after radiation, the stimulated parotid saliva flow rates were dramatically decreased by radiation alone and significantly recovered by the intra-parotid transfer of the Shh but not the GFP gene (Figure 1B). Consistently, in parotids at 20 weeks after radiation, the level of AQP5 protein and areas of AQP5^+^ acinar structures were greatly reduced by radiation and preserved by Shh but not GFP treatment (Figure 1C–E). Notably, in parotids at 1 or 5 weeks after radiation, levels of Gli1 and Ptch1, two target genes of Hedgehog signaling, were significantly decreased, whereas the transfer of the Shh but not the GFP gene at 4 weeks after radiation restored their expression at 5 weeks (Figure 1F,G). These data indicate that Hedgehog signaling in pig parotids is inhibited by radiation and restored by Shh gene transfer and confirm our previous report that transient activation of Hedgehog signaling rescues the function of radiated salivary glands. 

### 2.2. Hedgehog Receptor Ptch1 Is Mainly Expressed in Parotid Macrophages That Are Decreased by Radiation and Recovered by Transient Hedgehog Activation

The type of pig parotids cells that directly mediate the rescue of salivary function after radiation by transient Hedgehog activation remains unclear. Hedgehog signaling is triggered by the binding of Hedgehog ligands to their receptor Ptch1 on the cell membrane. Therefore, we examined the expression pattern of Ptch1 in the parotid glands of a healthy miniature pig using immunofluorescent staining. Ptch1 protein is mainly detected in cells with dendrites adjacent to ductal structures (Figure 2A), a typical morphology of salivary gland resident macrophages [15]. We tried several different mice and/or human antibodies of macrophage markers for immune staining the swine parotid sections, and only the antibody for ARG1, an M2 macrophage marker, succeeded. The double-immunofluorescent staining indicated that most Ptch1^+^ cells express ARG1 (Figure 2B), confirming that they are indeed macrophages. Radiation significantly decreased the expression of ARG1, pan-macrophage markers AIF1, F4/80 (ADGRE1) and ITGAX/CD11c, and resident macrophage-derived paracrine factors HGF and C1q [12,24] at protein and/or mRNA levels in parotids at 1 or 5 weeks, which was then reversed by the intra-parotid transfer of the Shh but not the GFP gene (Figure 2C–E). These data indicate that pig parotid resident macrophages are major Hedgehog-responsive cells that are sharply and persistently damaged by radiation but recovered by Hedgehog activation. 

### 2.3. Hedgehog Activation Repressed the Persistent Inflammation in Radiated Swine Parotids

In mouse models, radiation causes persistent inflammation of the salivary glands that contributes to irreversible dry mouth [7,8], while salivary-gland resident macrophages are considered anti-inflammatory [12,15]. To determine the effects of radiation and Hedgehog activation on the inflammation status of swine parotids, first, we examined the mRNA expression of multiple pro- vs. anti-inflammatory cytokines using qRT-PCR. As expected, radiation significantly increased the mRNA levels of pro-inflammatory IL6, TNF, and IFNG and decreased that of anti-inflammatory IL4, which was reversed by the transfer of the Shh but not the GFP gene (Figure 3A). Among these cytokines, IL6 is a major senescence-associated secretory phenotype (SASP) factor and is required for radiation-caused salivary gland hypofunction [7]. As indicated by immune staining, IL-6^+^ areas in swine parotid sections greatly increased at 1 and 5 weeks after radiation. This reaction is repressed by the Shh but not the GFP gene transfer (Figure 3B,C). The changes in IL-6 and TNF protein levels after radiation, with or without Hedgehog activation, were confirmed using Western blotting (Figure 3D,E). These data validate the presence of persistent inflammation caused by radiation in the swine salivary glands and suggest that the Hedgehog-mediated rescue of salivary gland function is related to the resolution of such persistent inflammation, likely through the recovery of anti-inflammatory resident macrophages. 

### 2.4. The Elevation of P53 Levels by Radiation Is Reversed by Hedgehog Activation 

We have reported that in radiation-induced cellular senescence in pig parotid glands, treatment with transient Hedgehog activation after radiation effectively inhibited cellular senescence and preserved the proliferation capacities of pig parotid cells [23]. P53 signaling mediates the cellular senescence and apoptosis caused by radiation in various tissues, including mouse salivary glands [7,25]. Therefore, we examined the levels of P53 protein and mRNA in pig parotids using Western blotting and qRT-PCR. As expected, radiation significantly increased P53 expression at both the protein and mRNA levels, while the intra-parotid transfer of the Shh but not the GFP gene repressed such increases (Figure 4A–C). The inhibition of P53 upregulation by Shh is likely due to both the direct activation of Mdm2 in Hedgehog-responsive cells [26] and the indirect relief of DNA damage stresses in other cells. For instance, Shh gene transfer after radiation restored the expression of HGF and C1q (Figure 2E) that can promote DNA repair [27], phagocytosis of dead or senescent cells, and the secretion of anti-inflammatory factors [28,29,30] that together resolve inflammation that propagates DNA damage and senescence [31].

### 2.5. VEGF Level Is Decreased by Radiation but Restored by Hedgehog Activation 

Vascular toxicity is another major mechanism of dry mouth caused by radiation. This mechanism was effectively prevented using VEGF (VEGFA) gene therapy [32]. Besides promoting angiogenesis, VEGF also inhibits cellular senescence in endothelial and other types of cells [33,34]. We have confirmed that the transient activation of Hedgehog signaling in the parotids of pigs 4 weeks after radiation restored salivary function for up to 20 weeks by preserving microvessels [23]. In other tissues, VEGF expression is regulated by Hedgehog signaling, or HGF, and C1q derived from Hedgehog-responsive macrophages [35,36,37]. Therefore, we examined VEGF expression in our swine parotid samples. Western blot analysis indicated that the VEGF protein level was significantly decreased by radiation at 1 and 5 weeks and restored by the intra-parotid transfer of the Shh but not the GFP gene at 4 weeks after radiation (Figure 5A,B). Immune staining of parotid sections confirmed the Western blotting data (Figure 5C,D). However, in samples without Shh treatment, the VEGF signal was mainly present in epithelial cells, including ductal and acinar structures, while in the Shh treated samples, the VEGF signal was restored in epithelial cells and emerged in stromal cells adjacent to ductal structures (Figure 5C). These expression patterns suggest that in radiated swine parotids, Hedgehog activation might directly induce VEGF expression in Hedgehog-responsive macrophages and restore VEGF expression in epithelial cells through macrophage-derived paracrine factors.

## 3. Discussion

Current research on the radiation injury of salivary glands focuses on epithelial cells, endothelial cells, and parasympathetic innervation, while the contributions of immune cells, especially the most abundant resident macrophages, are largely overlooked [38]. Our findings [12] initiate a major paradigm shift focusing on resident macrophages and their interactions with other cells.

Moreover, macrophages can be safely manipulated in cancer patients [39] and efficiently targeted by gene therapy or nano-medicine due to their high phagocytic activity [40], distinguishing macrophages as feasible therapeutic targets in cancer patients. 

However, Hedgehog activation appears not to be a clinically feasible approach to recover salivary-gland resident macrophages. First, the expression levels of Hedgehog receptors and mediators in salivary glands are low even before radiation and greatly decreased by radiation (Figure 1). Second, the doses of Hedgehog agonists in cancer patients are limited due to potential pro-tumor risks [41]. In other tissues, the maintenance of resident macrophages relies on local macrophage growth factors, including Csf1/IL34, Csf2 and/or Cx3cl1 [42]. Some of these growth factors are safe to use in cancer patients [39]. Therefore, to develop clinically feasible approaches for macrophage-based therapies for radiation damage of the salivary glands, our future work needs to explore the roles and therapeutic potentials of these macrophage growth factors and macrophage-derived pro-regenerative factors such as C1q and HGF.

One major limitation of our study using the pig model is the lack of reliable antibodies for flow cytometry and immune staining analysis of resident vs. infiltrating macrophages. For instance, the widely used mouse infiltrating macrophage marker Ly6c is not present in pigs. Recent advancements in single-cell RNA sequencing (scRNA-seq) techniques revealed some potential markers of resident vs. infiltrating macrophages in pigs [24]. We hope to use these markers and scRNA-seq in our future studies to deepen our understanding of swine resident macrophages in radiation damage and regeneration. 

Besides harming salivary glands, radiotherapy also causes long-lasting damage to many other healthy organs, such as the lungs and rectum, limiting radiation doses and greatly compromising the patient’s quality of life [43]. In many of these organs, resident macrophages are mainly derived from embryonic progenitors essential for functional regeneration and sensitive to radiation [42]. Therefore, recovering resident macrophages is likely a feasible strategy for alleviating radiation damage to these healthy organs as well.

Taken together, we found that in a clinically relevant swine model, radiation severely damaged the salivary gland resident macrophages identified as major Hedgehog-responsive cells, while transient Hedgehog activation after radiation recovered these cells, repressed cellular senescence and persistent inflammation, and restored salivary gland function. These data suggest that the loss of SG-resident macrophages is involved in the pathogenesis of human salivary gland hypofunction caused by radiation therapy for HNC, and these cells and their pro-regenerative products are promising targets to prevent or reverse this adverse effect.

## 4. Materials and Methods

### 4.1. Animals 

All animal work was approved by the Institutional Animal Care and Use Committee of Capital Medical University (AEEI-2015-089). BA-MA male miniature pigs, around 8 months old and weighing 25–35 kg, were purchased from the Institute of Animal Science of the Chinese Agriculture University (Beijing, China). Animals were housed and fed under conventional conditions and randomized into each treatment group. 

### 4.2. Irradiation 

The parotid target area of the miniature pigs was determined by axial computerized tomographic scans using a three-dimensional treatment planning system (Pinnacle3, version 7.6; ADAC Inc., Concord, CA, USA) as reported [23]. Animals were anesthetized with Ketamine and Xylazine and received single-dose irradiation of 20 Gy using image-guided radiation therapy (IGRT) technology. The radiation therapy plan was 6 mV of photon energy at 3.2 Gy/min using an Elekta Synergy accelerator (Elekta AB, Stockholm, Sweden). 

### 4.3. Gene Transfer and Saliva Collection

Adenoviral vectors encoding GFP or Rat Shh (Ad-GFP or Ad-Shh, Applied Biological Materials Inc., Richmond, Canada) were prepared and delivered into the parotids at 4 weeks after radiation therapy, as reported [23]. At 20 weeks after radiation, the parotid saliva samples were collected using a Carlson–Crittenden cup with a connected mechanical vacuum device (Shanghai ZhangDong Medical, Shanghai, China). The salivary flow rate was determined by the total parotid saliva collected in 10 min.

### 4.4. Histological and Immunohistochemical Analyses 

Parotids were collected at 1, 5 or 20 weeks after radiation therapy and fixed in 4% paraformaldehyde for histological or immunohistochemical detection. Samples were dehydrated with gradient ethanol and embedded in paraffin or frozen to obtain sections. Paraffin sections and frozen sections were subjected to immunohistochemistry (IHC) and immunofluorescent (IF) staining, respectively. These sections were incubated overnight at 4 °C with primary antibodies against AQP5 (1:200, ab78486), Ptch1 (1:200, ab51983), IL-6 (1:200, ab6672) (these 3 antibodies are from Abcam, Boston, MA, USA), ARG1 (1:100, A4923, Abclonal, Woburn, MA, USA), or VEGF (1:100, bs-1313R, Bioss, Woburn, USA). After being washed with PBS, appropriate secondary antibodies were incubated for 30 min at room temperature, and nuclei were counter-stained with DAPI for IF staining. The stained sections were imaged and quantified by investigators blinded to the treatments carried out in the experiment. At least two sections from 3 independent parotids were randomly chosen, and 2 random 200× fields of each section were imaged for quantification (n = 2 × 3 × 2 = 12). 

### 4.5. Quantitative RT-PCR and Western Blotting 

Total RNA extraction, reverse transcription, and quantitative PCR (qPCR) were performed as reported [44,45]. Primers for swine *GAPDH*, *C1qA*, *TNF-α*, *IFN-γ*, *IL-4*, *IL-6*, *P53*, *AIF1*, *ADGRE 1*, *ITGAX*, *HGF*, and *ARG 1* were designed using Primer3 software (http://bioinfo.ut.ee/primer3/) and listed in Appendix A. For protein extraction, fresh parotid samples were homogenized in T-PER reagent containing protease inhibitors (Pierce, WA, USA). The concentration of protein was quantified using BCA, and the loading quantity was 25 μg per lane. Western blotting was done using primary antibodies against AQP5 (1:5000, ab78486), Ptch1 (1:5000, ab51983), TNF-α (1:5000, ab1793), IL-6 (1:200, ab6672), P53 (1:500, ab154036) (these 5 antibodies are from Abcam, Boston, MA, USA), Gli1 (1:1000, bs-1206R), F4/80 (1:1000, bs-7058R), AIF1 (1:1000, bs-1363R), and VEGF (1:100, bs-1313R) (these 4 antibodies are from Bioss, Woburn, MA, USA). All PCR and Western blot analyses were repeated at least three times. 

### 4.6. Statistical Analyses 

All quantified data were analyzed using one-way ANOVA followed by Tukey’s multiple-comparison test in which SPSS 22.0. *p* < 0.05 was considered significant. Figures were generated using GraphPad Prism 6.0 software and presented as mean ± SD. 

## Figures and Tables

**Figure 1 ijms-22-13493-f001:**
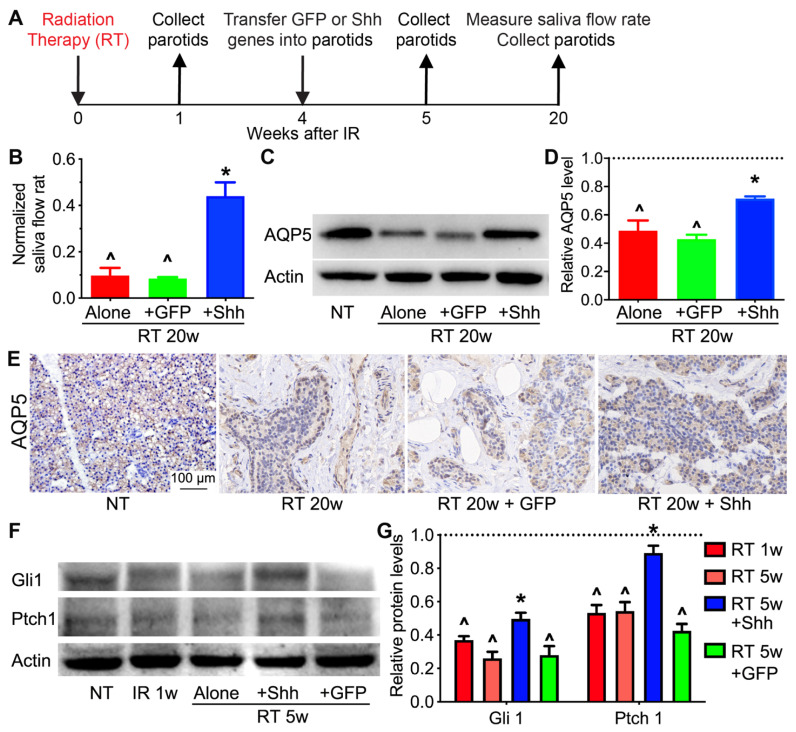
Effects of radiation and Shh gene transfer on parotid function and Hedgehog target gene expression. (**A**) Swine parotids were treated with radiation therapy (RT) on one side, and some of them were transferred with the Shh or GFP gene 4 weeks after RT. Parotids were collected at 1, 5, and 20 weeks after radiation. (**B**) At 20 weeks (20 w) after RT, the pilocarpine-stimulated parotid saliva flow rates were measured on the radiated and non-treated (NT) sides, respectively, and normalized to the NT rates of the same animal. (**C**–**E**) The AQP5 protein levels in untreated parotids or at 20 weeks after RT, with or without transfer of the Shh or GFP gene, were examined using Western blotting (**C**,**D**) and then confirmed through immunohistochemistry staining (**E**). (**F**,**G**) In parotids collected at 1 or 5 weeks after radiation, with or without transfer of the Shh or GFP gene, the protein levels of Hedgehog target genes Gli1 and Ptch1 were examined using Western blotting and normalized to that of Actin. N = 3, ^: *p* < 0.05 vs. NT, *: *p* < 0.05 vs. RT 5w and RT 5w + GFP.

**Figure 2 ijms-22-13493-f002:**
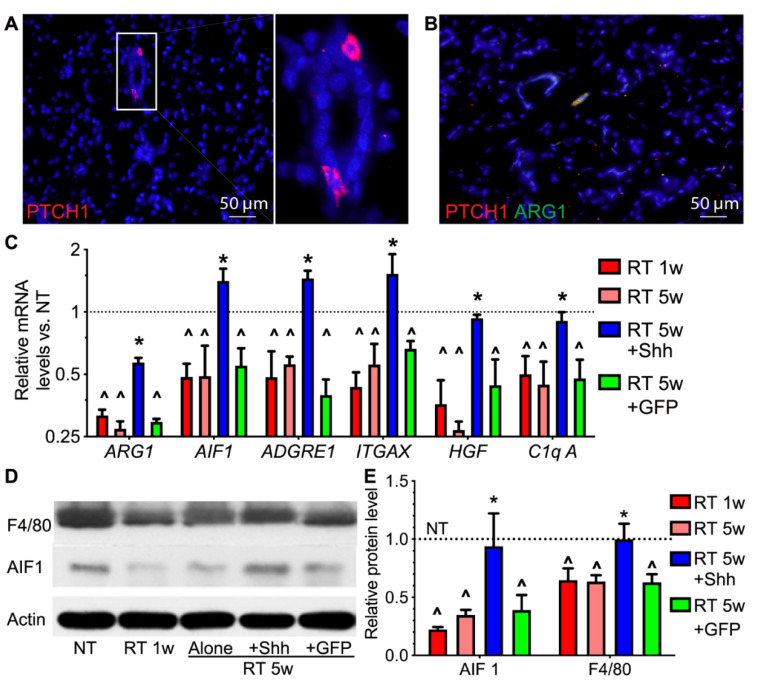
Effects of radiation and Shh gene transfer on the expression of macrophage markers in swine parotids. (**A**) In non-treated (NT) swine parotids, the expression of the Hedgehog receptor PTCH1 was mainly found in cells with the typical morphology of macrophages. (**B**) PTCH1 positive (red) cells express the M2 macrophage marker ARG1 (green). (**C**) The mRNA levels of ARG1 and other macrophage markers in NT and radiated parotids with or without the transfer of the Shh or GFP gene were examined with qRT-PCR. (**D**,**E**) The protein levels of F4/80 and AIF1 were examined using Western blotting and normalized to that of Actin. N = 3, ^: *p* < 0.05 vs. NT, *: *p* < 0.05 vs. RT 5w and RT 5w + GFP.

**Figure 3 ijms-22-13493-f003:**
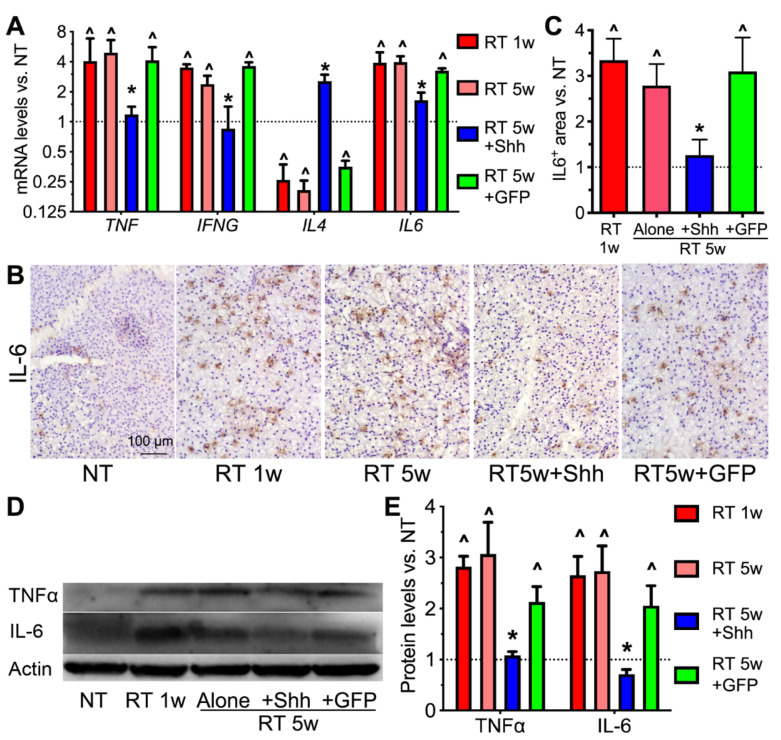
Effects of radiation and Shh gene transfer on the expression of inflammation-regulating cytokines. (**A**) The mRNA levels of pro- and anti-inflammatory cytokines TNF, IFNG, IL6 and IL4 were examined using qRT-PCR in NT and radiated parotids with or without transfer of the Shh or GFP gene. (**B**,**C**) The expression of IL-6 in parotid sections was examined using immunohistochemistry and quantified. (**D**,**E**) Protein levels of IL-6 and TNFα were examined using Western blotting and normalized to that of Actin. N = 3, ^: *p* < 0.05 vs. NT, *: *p* < 0.05 vs. RT 5w and RT 5w + GFP.

**Figure 4 ijms-22-13493-f004:**
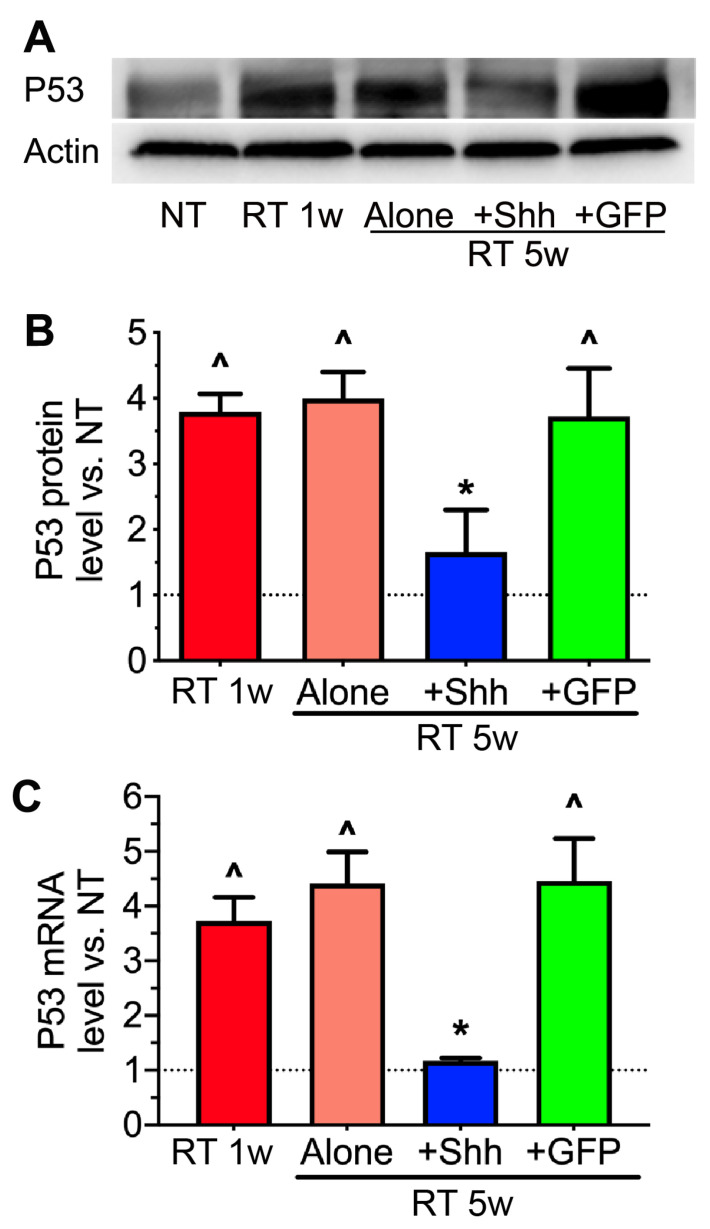
Effects of radiation and Shh gene transfer on the expression of P53. (**A**,**B**) The protein level of p53 in swine parotids was examined using Western blotting and normalized to that of Actin. (**C**) p53 mRNA level was examined using qRT-PCR. N = 3, ^: *p* < 0.05 vs. NT, *: *p* < 0.05 vs. RT 5w and RT 5w + GFP.

**Figure 5 ijms-22-13493-f005:**
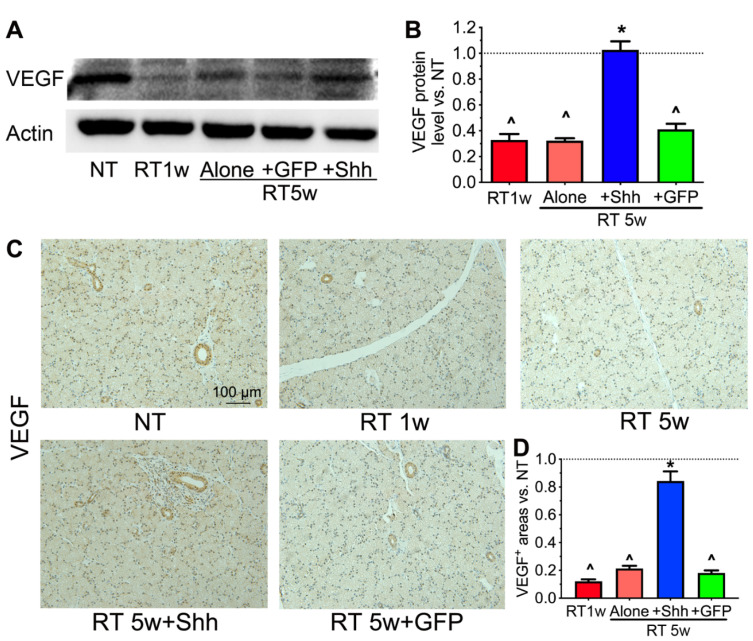
Effects of radiation and Shh gene transfer on the expression of VEGF. (**A**,**B**) The protein level of VEGF in swine parotids was examined using Western blotting and normalized to that of Actin. (**C**,**D**) The expression of VEGF in parotid sections was examined using immunohistochemistry and quantified. N = 3, ^: *p* < 0.05 vs. NT, *: *p* < 0.05 vs. RT 5 w and RT 5 w + GFP.

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
