# Peer review of "Transient Activation of Hedgehog Signaling Inhibits Cellular Senescence and Inflammation in Radiated Swine Salivary Glands through Preserving Resident Macrophages"

_ijms, 2021, doi:10.3390/ijms222413493_

Round 1

Reviewer 1 Report

This manuscript shows that the Hedgehog signaling pathway in pig parotids is inhibited by radiation and restored by Shh gene transfer thereby leading to preservation of salivary gland function. These studies are significant as they confirm previous report by the same group indicating that transient activation of Hedgehog signaling rescues function in irradiated salivary glands. Another important take home message of this paper is that pig parotid resident macrophages are major Hedgehog-responsive cells and persistently damaged by radiation. Conversely, Hedgehog activation in this model blocks inflammation, decreases p53 levels and improves VEGF levels all of which promote salivary gland function. Overall, the manuscript is well-written, the results are well presented, and the conclusions are well supported by the results. I believe the manuscript should be published pending minor revisions as detailed below:

  1. The last sentence in the abstract should be about how Hedgehog activation leads to restoration of function in irradiated salivary glands.
  2. Irradiation is the process by which an object is exposed to radiation. As such, the term irradiation should be replaced by radiation therapy throughout the manuscript.
  3. Salivary hypofunction is not the same as xerostomia. Particularly, hypofunction is the reduction of saliva flow rates while xerostomia refers to a symptom which is the sensation of dry mouth. This should be corrected.
  4. Fig. 1D, 2A, 3B and 5C are missing the scale bars
  5. Fig. 1E Western blot images for Gl1 and Ptch1 are blurry and should be replaced
  6. Fig. 3D, Western blot for interleukin 6 is cut off and should be replaced
  7. Fig. 5A, Western blot for VEGF is blurry and should be replaced for a sharper image

Author Response

We sincerely thank the reviewers for their constructive critiques of our previous version of manuscript and revised our manuscript accordingly. 

  1. The last sentence in the abstract should be about how Hedgehog activation leads to restoration of function in irradiated salivary glands.

Response: We revise that to “transient Hedgehog activation restores the function of irradiated salivary glands through the recovery of resident macrophages and the consequent inhibition of cellular senescence and inflammation”.

  1. Irradiation is the process by which an object is exposed to radiation. As such, the term irradiation should be replaced by radiation therapy throughout the manuscript.

Response: The term irradiation is replaced by radiation therapy or radiation throughout the manuscript.

  1. Salivary hypofunction is not the same as xerostomia. Particularly, hypofunction is the reduction of saliva flow rates while xerostomia refers to a symptom which is the sensation of dry mouth. This should be corrected.

Response: Corrected.

  1. 1D, 2A, 3B and 5C are missing the scale bars

Response: Scale bars are added.

  1. 1E Western blot images for Gl1 and Ptch1 are blurry and should be replaced

Response: Replaced.

  1. 3D, Western blot for interleukin 6 is cut off and should be replaced

Response: Replaced.

  1. 5A, Western blot for VEGF is blurry and should be replaced for a sharper image

Response: Replaced.

Reviewer 2 Report

In this article by Liang Hu et colleagues, the authors report that in swine parotid glands Hedgehog receptor PTCH1 is mainly expressed in macrophages, and levels of PTCH1 and multiple macrophage markers are significantly decreased by radiation but recovered by transient Hedgehog activation.

Comments and suggestions:

  1. A diagram with a study design will be welcomed for the readers.
  2. Introduction section: add more data about human salivary glands and radiotherapy for head and neck cancers. Only one sentence is not enough.
  3. Material and methods: I suggest adding as separate subsection Animals, Irradiation
  4. What significance for humans does this MS have?

Consider revision accordingly.

Author Response

We sincerely thank the reviewers for their constructive critiques of our previous version of manuscript and revised our manuscript accordingly. 

  1. A diagram with a study design will be welcomed for the readers.

Response:  A diagram with a study design is added as Figure 1A.

  1. Introduction section: add more data about human salivary glands and radiotherapy for head and neck cancers. Only one sentence is not enough.

Response:  Several sentences and two recent references are added on human salivary glands and their irreversible damage by radiotherapy.

  1. Material and methods: I suggest adding as separate subsection Animals, Irradiation

Response:  Added.

  1. What significance for humans does this MS have?

Response: Our data suggest that the loss of SG-resident macrophages are involved in the pathogenesis of human salivary gland hypofunction caused by radiation therapy for HNC, and these cells and their pro-regenerative products are promising targets to prevent or reverse this adverse effect. We replace the last sentence of Discussion with above significance statement.